# The Potential Role of School Citizen Science Programs in Infectious Disease Surveillance: A Critical Review

**DOI:** 10.3390/ijerph18137019

**Published:** 2021-06-30

**Authors:** Ayat Abourashed, Laura Doornekamp, Santi Escartin, Constantianus J. M. Koenraadt, Maarten Schrama, Marlies Wagener, Frederic Bartumeus, Eric C. M. van Gorp

**Affiliations:** 1Department of Viroscience, Erasmus Medical Center Rotterdam, 3015GD Rotterdam, The Netherlands; l.doornekamp@erasmusmc.nl (L.D.); m.n.wagener@hr.nl (M.W.); e.vangorp@erasmusmc.nl (E.C.M.v.G.); 2Center of Expertise Innovations in Care, Rotterdam University of Applied Sciences, 3015EK Rotterdam, The Netherlands; 3Centre d’Estudis Avançats de Blanes (CEAB-CSIC), 17300 Blanes, Spain; santi.escartin@ceab.csic.es; 4Laboratory of Entomology, Wageningen University & Research, 6708PB Wageningen, The Netherlands; sander.koenraadt@wur.nl (C.J.M.K.); fbartu@ceab.csic.es (F.B.); 5Institute of Environmental Sciences, Leiden University, 2300RA Leiden, The Netherlands; m.j.j.schrama@cml.leidenuniv.nl; 6Centre de Recerca Ecològica i Aplicacions Forestals (CREAF), 08193 Cerdanyola del Vallès, Spain; 7Institució Catalana de Recerca i Estudis Avançats (ICREA), 08010 Barcelona, Spain

**Keywords:** citizen science, education, infectious diseases, life sciences, public health, schools, surveillance

## Abstract

Public involvement in science has allowed researchers to collect large-scale and real-time data and also engage citizens, so researchers are adopting citizen science (CS) in many areas. One promising appeal is student participation in CS school programs. In this literature review, we aimed to investigate which school CS programs exist in the areas of (applied) life sciences and if any projects target infectious disease surveillance. This review’s objectives are to determine success factors in terms of data quality and student engagement. After a comprehensive search in biomedical and social databases, we found 23 projects. None of the projects found focused on infectious disease surveillance, and the majority centered around species biodiversity. While a few projects had issues with data quality, simplifying the protocol or allowing students to resubmit data made the data collected more usable. Overall, students at different educational levels and disciplines were able to collect usable data that was comparable to expert data and had positive learning experiences. In this review, we have identified limitations and gaps in reported CS school projects and provided recommendations for establishing future programs. This review shows the value of using CS in collaboration with traditional research techniques to advance future science and increasingly engage communities.

## 1. Introduction

### 1.1. The Current Dilemma of Infectious Diseases and the Need for Early Detection Strategies

Emerging infectious diseases pose a clear burden for public health security and progression [1,2]. Recent epidemics such as Ebola, Zika, H1N1 and SARS-CoV-1 viruses have emerged and re-emerged in all parts of the world [3,4]. It is widely agreed that environmental changes, globalization, social-economic conflicts are important underlying factors for infectious disease emergence and transmission [5]. The COVID-19 pandemic underscored the weaknesses of current surveillance systems and the demand for better early detection strategies worldwide [6]. Traditional detection methods are effective but could be improved: many infections remain undetected due to poor diagnostic tools, leaving populations untreated and delays in identification and detection of infectious diseases can lead to serious human and economic repercussions [7]. The COVID-19 pandemic is a turning point that lends researchers the opportunity to develop enhanced early detection approaches.

### 1.2. Citizen Science as a Research Tool

In the last few decades, utilizing citizen science (CS) methodologies and data has become more widespread in research. Nowadays, with the need for comprehensive data sets that cover wide spatial and temporal scales, citizen scientists have emerged as an advantageous asset who have helped speed up data collection and diversify datasets for researchers [8,9]. As defined in 1995 by the Cornell Lab of Ornithology, CS involves any nonprofessionals in collecting data, processing data and sometimes even creating study designs [10]. While CS publications are becoming more now, the concept of involving the public in scientific processes is far from new; in the mid-1800s, poet and philosopher Henry David Thoreau recorded over 500 plant species in his living environment and his observations have been used to determine how global warming has affected flowering times over the years [11]. Additionally, the ideas of CS are not new and are rooted in the educational theory of participatory action research (PAR) [12]. PAR combines theory and practice by having researchers and participants collaborate to study a problem and determine how to create a solution; PAR has been implemented as early as the 1950s in school settings to enhance scientific literacy and is still used to this day [13], just as CS is now being employed. Over the years, there have been many collaborative projects between community members and professional scientists, leading to many different definitions of CS. Wilderman’s taxonomy proposed to classify these collaborations into different themes based on volunteer participation in the following activities: problem definition, study design, sample/data collection, data analysis and data interpretation [14].

By engaging community members through CS, not only do researchers gain invaluable data, but the public also gains scientific knowledge. Community participation leads to an increase in scientific literacy, another instrumental aim of CS program implementation [15,16]. For individual participants, CS can offer personal fulfilment by achieving personal learning goals, aiding in scientific discoveries or even just having fun [17,18]. Many fields have already benefited in obtaining usable data by employing a CS model: archaeology [19], astronomy [20], biochemistry [9], ecology [21], environmental monitoring [22], geography [23], taxonomy [24] and oceanography [25].

### 1.3. Students as Citizen Scientists

Although the majority of volunteers in CS projects have been highly educated adult men and women [26], some CS projects aim to work with younger volunteers. Advocates for CS encourage schools to incorporate CS plans in their curriculums—as is being done in Spain—to increase student interest, curiosity and knowledge in scientific fields [27]. In a 2017 study, researchers assessed youth participation in three CS projects for collecting conservation data. For youth that were very motivated, they really focused on ensuring the data they collected was high quality, and they saw a clear link in data quality for establishing environmental community actions. These CS projects had a positive effect on conservation research and management, and they also educated and activated younger communities [28]. The benefit to these more participatory approaches in education were shown to be evident in a study spanning elementary, junior high and high school years over a six-year period conducted in the United States. Students became ‘co-researchers’ in a project and had to complete typical research activities such as creating questions, leading interviews and analyzing data; by the end, they had established their own motivations, personalized their studies and added to scholarly knowledge [29]. Thus, bridging the gap between students, teachers and researchers by integrating CS into classrooms enhances scientific awareness.

### 1.4. Citizen Science as a Method for Infectious Disease Surveillance

An area that could benefit from CS is infectious disease surveillance. For example, cases of influenza go unreported yearly, with figures from hospitals and general practitioners underestimating the true prevalence of influenza. Integrating CS reporting of influenza cases, both in humans and animals, illuminates on not only disease prevalence but on transmission dynamics as well [30]. Besides influenza, to our knowledge, there are no programs (community or even school-level) geared at utilizing CS to target infectious disease surveillance within the public before 2020. However, with the COVID-19 pandemic, researchers have begun to use CS to collect data. At San Diego State University, doctoral students have created testing kits for citizen scientists to use to take environmental coronavirus samples in public spaces. With this data, scientists will create models to predict transmission risk of COVID-19 from the environment [31]. CS is a real-time advantageous tool to help combat future epidemics and pandemics. Schools are dynamic meeting points where a large section of the population interacts on a daily basis. Large social mixing and social interactions between students in schools is a known primary mode of disease transmission [32,33,34], thus, illustrating the great potential in implementing CS school projects aimed at infectious disease surveillance in school environments.

While CS programs are now becoming more prevalent in various scientific fields, such as environmental sciences and ecology, the extent of infectious disease surveillance CS programs at school levels is unexplored. In this critical review, we investigate what programs in life sciences and applied life sciences have incorporated CS into various classroom settings, and the impact CS has on two outcome measures: data quality and student engagement. This results in a comprehensive overview of the benefits and challenges to CS in youth and how CS can contribute to infectious disease surveillance.

## 2. Materials and Methods

For this critical review, we developed a search strategy and applied inclusion/exclusion criteria to construct a CS in education inventory.

### 2.1. Search Strategy

We gathered peer-reviewed literature searched from PubMed, Education Resources Information Center (ERIC), SocINDEX (EBSCO) and PsycINFO. For the search, we used (a variation of) the following terms: (1) “citizen science” and “education” or (2) “citizen science” and “school”. For the databases, queries were restricted to titles, abstracts and full texts written only in English. To be as thorough as possible and to ensure inclusion of projects that might lend chronological perspective, searches were not restricted by date. This review reflects the literature published prior to the fourth of August 2020.

### 2.2. Inclusion and Exclusion Criteria

A four-step screening process was applied to the articles. First, to determine CS relevance for the review, we applied Wilderman’s taxonomy (problem definition, study design, sample/data collection, data analysis and data interpretation) as defined earlier [14]. We applied Wilderman’s taxonomy because it helps depict the primary research goal of the review which is to provide an overview of the available school CS programs and their success in data collection efficacy. The secondary goal is to report on student engagement since not all CS projects aim to study social outcomes. For this review, we included a project if the participants were engaged in at least one of the before-mentioned research activities but with one additional condition: If the participants collected data on themselves (e.g., oral rinse sample), then the participants needed to be involved in one other research activity in order for the project to be considered relevant and included. Incorporating this caveat ensured that projects included did not focus on citizens as the research subjects themselves.

Second, the projects had to be conducted in an educational institute so that the volunteers were students. We included students from primary, secondary, post-secondary/tertiary (e.g., technical/community college) and university (bachelor’s and master’s) levels. The educational levels are defined by the International Standard Classification of Education (ISCED) [35].

Third, only projects that had been completed were included. Any article that focused on an ongoing project was excluded in order to be able to discuss the impact of the project results. We also excluded any CS project involving schools not aimed at achieving any kind of scientific publication, such as projects falling more into PAR.

Lastly, we restricted this review to only include articles related to life sciences, applied life sciences and derived concepts (such as public health) and further divided the articles into seven categories based on their main CS research theme, the scope of the journal, and their associated manuscript keywords: Ecology, Genomics & Genetics, Biological Conservation, Microbiology, Public Health, Environmental Health and Botany. Articles related to natural sciences such as physics and chemistry were excluded.

### 2.3. Outcome Measures

The results of the search were evaluated based on two outcome measures or indicators: (1) data quality and (2) student engagement. Data quality was based on data validity, reliability or usability for expert research as determined by the manuscript’s results. We defined student engagement as students’ reactions to project participation such as knowledge, interest, confidence or motivation and was evaluated in terms of interviews or surveys. Data quality and student engagement were then scored for each project according to the following scale: 0 = not mentioned, 1 = low, 2 = moderate or 3 = high.

## 3. Results

The database searches yielded 4623 articles. After applying the inclusion and exclusion criteria, 23 projects were eligible for the literature review inventory (Figure 1) and divided into their corresponding research fields. Short descriptions of the studies are given in Table 1. For each project, Wilderman’s taxonomy and outcome measures are reported in Table 2. The average scores for data quality and student engagement in each CS field are shown in Figure 2.

Of the 23 projects, the majority took place in North America (specifically the United States) and Europe (n = 10 and 8, respectively). The rest were conducted in Australia/Oceania (n = 3) and South America (n = 2). Based on education level, most of the projects involved secondary school students (n = 15). The remaining projects were conducted at four primary educational schools and four universities, and one project did not mention which educational levels participated (Table 1).

### 3.1. Ecology

The topics of five of the 23 selected projects were based on ecological research. Two of the projects focused on students collecting and identifying species in the United States and Australia [36,37]. In the United States, students in secondary years (grades 6 through 12) were able to collect high quality data in identifying already established species that was not statistically different to expert results [36]. In Australia, university students were not assessed on the quality of their data, but the surveys showed that students’ environmental engagement increased after collecting and analyzing data [37].

In Brazil, ecological CS activities were completed by students in Brazilian primary and secondary schools. Using simplified protocols, the student-collected data showed similar patterns to that of the verified samples, exposing habitat and water quality degradation of streams [38]. Another project had participating primary school students to assess biodiversity of bumblebees based on landscape [39]. While data quality collected by the Brazilian primary students was high [38], data quality was generally not very high in terms of bumblebee color identification by the primary school students [39]. Also, for both projects, there was no mention of how students were engaged in the whole experience [38,39].

A study by Tarter et al. assessed high school student-monitored *Aedes aegypti* dynamics in Arizona. While the students did not find many new areas of mosquito activity, researchers claimed students to be successful at completing mosquito surveillance but did not address data quality. Recruitment and sustainability of the project was difficult over the three-year study period. While participation in the program was higher in the first two years, teachers noted that the students had moderate or high interest in the project and student engagement increased in the final year [40].

### 3.2. Genomics and Genetics

Five of the 23 selected projects were based in the fields of genomics and genetics with four specifically involving DNA barcoding databases with secondary school students (Table 1). The DNA barcoding projects were primarily conducted by high school students in the United States and Australia whose results either generated new sequences of marine species to add to the database to assess biodiversity or confirmed the accuracy of marketplace seafood labelling. For the four studies, researchers noted high data quality and high student engagement, with the exception of the Marizzi et al. study, which only recorded high data quality [41,42,43,44]. At the University of Oviedo in Spain, students produced many successful PCRs of DNA markers in seafood products in various classes in terms of data quality. Students were more motivated and evaluated the courses better when working with real samples sourced from their local markets [45].

### 3.3. Biological Conservation

Of the 23 selected projects, four focused on biological conservation. In Chile and Germany, three studies focused on litter pollution [46,47,48]. Using the CS protocol from Chile [46], a 2019 German study evaluated marine debris density and composition. Instead of secondary school students, primary school students collected debris data along German coastlines. The German students produced valid and reliable data just like the Chilean predecessors [46,47]. Another German study by Kiessling et al. had many issues with student-collected datasets due to missing photos and information and much litter misidentification during the 2016 program. However, the protocol was changed to be less complex, leading to much higher quality data that was usable for experts in 2017 [48]. Unlike the 2013 Chilean study, both German studies did not elaborate on student engagement and did not specify the students’ educational level [46,47,48].

Schools over the course of 18 years in France, collected and prepared water samples for scientists to analyze nutrients in rivers and agricultural water bodies. Overall, this CS initiative produced high quality nutrient seasonality data and concluded that incorporating participatory water monitoring could help maintain community water quality. There was no mention of student engagement [49].

### 3.4. Microbiology

Four of the 23 CS programs concentrated on microbiological techniques and themes. Students in three of the studies isolated bacterial samples [50,51,52], with two of the projects using the Small World Initiative (SWI) protocol [50,51]. At Bowling Green State University, students isolated bacteria, but Davis et al. did not comment on data quality and student engagement. However, they did conclude that students can contribute to important research through CS [40]. In Spain, the SWI protocol was adjusted for high school students. Unlike the Davis et al. study, data quality and student were assessed, with 23 positive hits found by the students and overall positive participation from the students, respectively [51]. At North Carolina State University, there were no direct results about student engagement, but Riley et al. noted that the project helped teach students about *Delftia* species that were sampled at different campus locations. While the accuracy of the diversity of the *Delftia* strains could not be determined, the sequencing results most closely matched the environments studied by Riley et al.

A study by Abe et al. compared microbial samples from pre-existing and newly fitted showerheads collected by high school students in Hawaii and Colorado, but scientists did not comment on the data quality and accepted the results. Students noted that the experience increased their confidence in outside school work, learning microbial techniques, acquiring knowledge on microbes that impact public health and enhanced critical thinking [53].

### 3.5. Public Health

Only two CS projects had a public health focus. The Youth Participatory Action Research (YPAR) 2.0 project aimed to develop new methods for high school students to reveal social inequalities based on food scarcity in a California neighborhood. After the data was validated and verified, the student-data exposed that East Oakland did not actually have 50 grocery stores but only three, with the rest being liquor stores. Participating youth felt empowered and more confident in linking knowledge of public health inequity to community action [54]. In the Walkinshaw et al. study, scientists tested the quality of student-collected water source photo data to identify water accessibility issues at schools. While students adhered to the protocol strictly for only 60% (n = 24) of the schools, 98% of the data from the schools (n = 39) was usable, and the data reliability averaged 0.95 interrater agreement across all measures. The majority of students found the experience to be valuable and even urged that the project would be made more challenging [55].

### 3.6. Environmental Health

Two of the CS projects aimed to answer questions related to environmental health with the help of secondary school students. In the United States, high school students deployed sensors to monitor air quality and created a website to monitor real-time air quality data, which agreed with federal reference monitoring data. High data quality was reached, mainly due to the open communication and feedback with teachers and students [56]. The FreshWater Watch in Ireland recruited high school students to test if they could measure five water quality parameters based on the Sustainable Development Goals (SDGs) (Indicator 6.3.2). Of the five parameters, researchers found that three of the five parameters (orthophosphate, nitrate and electrical conductivity) these citizen scientists collected were comparable to laboratory results. The remaining two parameters collected, pH and chemical oxygen demand, were not reliable. Scientists did not comment on student engagement. While the students were successful in collecting high-quality data for three parameters, the scientists acknowledged that the measurement kits provided should be tested for accuracy and precision [57].

### 3.7. Botany

In the United Kingdom, primary school students measured spring onions for researchers to combine with temperature measurements to create predictive growth models. Student-measured spring onion circumferences were closely matching to the expert-collected measurements. While there were some issues in data reporting, most of the data was valid. Student engagement was not reported [58].

**Table 1 ijerph-18-07019-t001:** Summary of school citizen science projects described in scientific literature.

Field	Author (Date) [Reference]	Description of Citizen Science Project	Education Levels	ISCED	Location	Project Timeline
Ecology	Cox, TE et al. (2012) [36]	Determined if students could identify and describe abundances and distributions of species in intertidal habitats accurately	Grades 6–12	Secondary Education	Hawaii, United States	2004–2007
	Roy, HE et al. (2016) [39]	Assessed the influence of landscape on the diversity and abundance of bumblebees	Primary school (7–11 years old)	Primary Education	United Kingdom	Not specified
	Mitchell, N et al. (2017) [37]	Reported on CS program allowing university students to record phenological information on indicator species occurring in Western Australia	University	Not specified	Perth and Albany, Australia	2011–2017
	França, JS et al. (2019) [38]	Monitored ecological quality of urban streams	Elementary, middle and high schools	Primary and Secondary Educations	Belo Horizonte, Brazil	2013–2017
	Tarter, KD et al. (2019) [40]	Assessed usability and feasibility of CS-monitored *Aedes aegypti* mosquitoes in Arizona	High Schools	Secondary Education	Arizona, United States	2015–2017
Genomics & Genetics	Santschi, L et al. (2013) [41]	Monitored marine specimens and assigned them to specific taxa for database records	Grades 11 and 12	Secondary Education	California, United States	Not specified
	Borrell, YJ et al. (2016) [45]	Developed genetics laboratory practices and measured the impact of food products to understand Food Control topic in university courses	University	Bachelor’s and Master’s	Asturias, Spain	2014–2015 Academic year
	Marizzi, C et al. (2018) [42]	Assessed biodiversity using DNA barcoding at Brooklyn’s Marine Park	High school	Secondary Education	New York, United States	2014–2015
	Chiovitti, A et al. (2019) [43]	Developed educational program in which students conducted research in DNA barcoding	Grades 11 and 12	Secondary Education	Victoria, Australia	2013–2014
	Mitchell, A et al. (2019) [44]	Conducted DNA extraction, isolation and amplification to generate preliminary scientific data on the accuracy of species labelling in marketplaces	High school	Secondary Education	Sydney, Australia	2015–2016
Biological Conservation	Hidalgo-Ruz, V et al. (2013) [46]	Assessed the distribution and abundance of small plastic debris on beaches	Middle and high schools (8–16 years old)	Secondary Education	Chile	October to November 2011
	Abbott, BW et al. (2018) [49]	Analyzed river nutrients in agricultural catchments	High school (second years)	Secondary Education	France	September 1998 to December 2015
	Honorato-Zimmer, D et al. (2019) [47]	Examined anthropogenic marine debris density and composition differences between Chile and Germany	Grades 5–12	Primary and Secondary Educations	Germany	Not mentioned
	Kiessling, T et al. (2019) [48]	Estimated litter quantity in rivers and identified litter material composition	Specific levels not mentioned	Not specified	Germany	September–November 2016 and May–July 2017
Microbiology	Abe, J et al. (2016) [53]	Compared the microbial diversity on pre-existing and on newly installed showerheads	High school	Secondary Education	Hawaii and Colorado, United States	9 months (specific year not mentioned)
	Davis, E et al. (2017) [50]	Isolated bacteria from local environments, characterized the strains, and assayed for antibiotic production	University	Not specified	Kentucky, United States	Fall 2015
	de Groot, PWJ et al. (2019) [51]	Modified Small World Initiative/Tiny Earth protocols to be done by students to improve and optimize isolating antibiotic-producing bacteria	High school	Secondary Education	Albacete, Spain	Not specified
	Riley, NG et al. (2020) [52]	Tested CS campus-wide microbial project to understand the diversity and distribution of bacterial genus *Delftia*	University	Not specified	North Carolina, United States	January to April (year not specified)
Public Health	Akom, A et al. (2016) [54]	Developed new technologies for students to visualize, validate and transform social inequalities based on food scarcity	High school	Secondary Education	California, United States	Summer 2011
	Walkinshaw, LP et al. (2019) [55]	Tested feasibility of students to collect high quality school water source photo data	High school	Secondary Education	United States	2016–2017
Environmental Health	Hyder, A et al. (2020) [56]	Described environmental health translational data analytics project with high school involvement	High school	Secondary Education	Ohio, United States	2016–2020
	Quinlivan, L (2020) [57]	Investigated if students could collect high quality data on a number of ambient water quality parameters associated with SDG Indicator 6.3.2	High school (16–17 years old)	Secondary Education	Kerry, Ireland	2019
Botany	Brestovitsky, A et al. (2019) [58]	Determined how spring onions develop in response to temperature	Primary school (9–11 years old)	Primary Education	United Kingdom	Over 2 weeks (year not mentioned)

ISCED: International Standard Classification of Education; CS: Citizen Science; SDG: Sustainable Development Goal.

**Table 2 ijerph-18-07019-t002:** Wilderman’s taxonomy and outcome measures of school citizen science projects.

Field	Author (Date) [Reference]	Wilderman’s Taxonomy	Data Quality	Student Engagement
Ecology	Cox, TE et al. (2012) [36]	Data Collection	Moderate	Not mentioned
	Roy, HE et al. (2016) [39]	Data Collection Data Interpretation	Low	Not mentioned
	Mitchell, N et al. (2017) [37]	Data Collection Data Analysis Data Interpretation	Not mentioned	High
	França, JS et al. (2019) [38]	Data Collection	High	Not mentioned
	Tarter, KD et al. (2019) [40]	Data Collection	Not mentioned	High
Genomics & Genetics	Santschi, L et al. (2013) [41]	Data Collection	High	High
	Borrell, YJ et al. (2016) [45]	Data Collection Data Analysis	Not mentioned	High
	Marizzi, C et al. (2018) [42]	Data Collection Data Analysis	High	Not mentioned
	Chiovitti, A et al. (2019) [43]	Data Analysis	High	High
	Mitchell, A et al. (2019) [44]	Data Analysis	High	High
Biological Conservation	Hidalgo-Ruz, V et al. (2013) [46]	Data Collection Data Analysis Data Interpretation	Moderate	High
	Abbott, BW et al. (2018) [49]	Data Collection	Low then high	Not mentioned
	Honorato-Zimmer, D et al. (2019) [47]	Data Collection Data Analysis Data Interpretation	High	Not mentioned
	Kiessling, T et al. (2019) [48]	Data Collection Data Analysis Data Interpretation	Moderate	Not mentioned
Microbiology	Abe, J et al. (2016) [53]	Data Collection Data Analysis Data Interpretation	Not mentioned	High
	Davis, E et al. (2017) [50]	Data Analysis Data Interpretation	Not mentioned	Not mentioned
	de Groot, PWJ et al. (2019) [51]	Data Analysis Data Interpretation	High	Moderate
	Riley, NG et al. (2020) [52]	Data Collection Data Analysis	High	Not mentioned
Public Health	Akom, A et al. (2016) [54]	Data Collection Data Analysis	High	High
	Walkinshaw, LP et al. (2019) [55]	Data Collection	High	High
Environmental Health	Hyder, A et al. (2020) [56]	Study Design Data Collection Data Analysis Data Interpretation	High	High
	Quinlivan, L (2020) [57]	Data Collection Data Analysis	Moderate	Not mentioned
Botany	Brestovitsky, A et al. (2019) [58]	Data Collection	Moderate	Not mentioned

## 4. Discussion

In this review, we described students’ participation in CS projects in recent years in several fields with a high proportion in ecology, microbiology and genetics and genomics. Data quality is generally moderate-to-high or high quality often occurring when protocols are explicit. Moreover, in the cases where it was measured, CS projects at schools appeared to have a very positive effect on student engagement. Data quality and student engagement are two first-order indicators that are simple enough to be minimally traceable and assessable across the literature. Data quality is a main concern in CS programs: the expectation is that citizens’ actions might lead to low accuracy and large sampling biases since CS data collection methods may not be as strict or expensive as traditional science [59,60]. Student engagement is a key educational indicator as is a clear reflection of the learning potential of CS. Using CS in school curriculums enhances knowledge and can complement teachers’ lesson plans [28,55,59].

Some of the school projects noted issues with data misidentification and misinterpretation such as distinguishing plastic and natural debris [46,47], recognizing bumblebee color groups [39] and determining the range for water quality results based on a colorimetric method [57]. Other studies had reporting issues due to missing or incomplete measurements, photographs or information, which led to rejection of datasets [48,55,58]. To account for these limitations, creating detailed yet simplified protocols and providing proper training [46,47,48], increases the quality of data collected by students and can match that of scientists [46]. It is also key to plan *a priori* the fitness of use of a CS program. Intentionally designing a program’s final goals and making decisions on whether it should be aligned to awareness and PAR-like goals or actual scientific gain and direct contribution to science [61] are all crucial to program design. In this context, previous studies have shown that proper training and adopting applicable protocols is crucial for citizens to collect expert-level data [62]. Students were also asked to resubmit data if information was missing, thereby guaranteeing usable data [55]. Indeed, CS data collected by students is usable, and implementing a sustainable program at the school level is feasible as shown in the Cox et al. and Mitchell et al. studies on species identification. As CS programs advance, choosing appropriate technologies will need to be considered to ensure reliable data collection [63] within the capabilities of school participants.

At the school level, most of (applied) life science CS projects have been implemented with success, but the majority are aimed at species biodiversity and pollution. Of the 23 programs included in this review, none truly concentrated on infectious disease surveillance. On a grander scale, CS projects usually do not focus so much on infectious disease surveillance and epidemiology, but primarily on ecology, environmental sciences, microbiology and geography. As mentioned before, a strategy similar to CS is ‘popular epidemiology’ [64]. Brown defines popular epidemiology as the process by which laypeople collaborate with scientific experts to collect data (usually environmental) that contribute to disease epidemiology [65]. The Great Arizona Mosquito Hunt pilot by Tarter et al. focused on *Aedes aegypti* mosquito surveillance, which falls under this popular epidemiology method but is also CS. However, vector surveillance through CS programs such as “Mosquito Alert” and “Muckenatlas” have been effective ways for scientists to link mosquito abundances to disease spread in Spain and Germany [60,66,67]. Recently, scientists have employed CS for COVID-19 surveillance. In the United Kingdom, using data from the “COVID Symptom Study” mobile application, scientists created predictive models that found that loss of smell is a predictor for COVID-positive tests and determined COVID-19 hotspots [68]. In the United States, the American Lung Association and universities collaborated to create another mobile-based COVID-19 CS study [69]. Thus, incorporating these types of CS platforms in school curriculums could be an initiative to promote indirect infectious disease surveillance and raise awareness among scholars.

There are many benefits for scientists, students and teachers to participate in CS projects. Firstly, CS allows scientists to fulfill their responsibilities to disseminate science: its importance to the public and its ability to disprove misinformation. Since many scientific projects are funded by the tax-paying public, people deserve to understand where their money is being invested and understand the significance of the research: also as a way to thank the public [70]. As infodemics become more prevalent [71], using CS as an educational tool can fight against misinformation. Secondly, involving schools in research projects provides more massive datasets that are otherwise impossible to compile [49]. A key advantage of these CS-collected datasets is their diversity in variables (e.g., locations and time points) that are otherwise hard to sample or missed with traditional research methods [40,47,72,73]. For instance, using CS as an additional monitoring tool can aid in achieving the SDGs [74]. Participatory water monitoring programs in Peru have provided data for watershed planning which compiles data related to SDG 6 of clean water and sanitation [75]. In the Philippines, community members collect information on health, poverty, nutrition, housing, education and disaster risk reduction to provide additional statistics for the Philippine Statistics Authority for 32 SDG indicators [76,77]. Thus, CS is complementary to active and routine scientific surveillance programs, rather than entirely replacing them.

In addition to providing more comprehensive databases, CS impacts scientific literacy and personal development and satisfaction. For many of these school CS programs, students and teachers alike showed initial and continued interest, which was evident in interviews and evaluations [41,78]. Studies show that including students in interactive research problems can improve classroom performance and retention [79]. Students gained a better understanding of specific topics and increased awareness of issues in their local areas [46]. Linking the students and the public to the scientific sphere through active community engagement creates a symbiotic relationship to solve community health issues and creates public empowerment [54,80]. CS studies that focused on population health, such as public space quality or food access, revealed new ways for refining urban planning by active volunteer involvement [81,82]. Incorporating more of these types of CS projects at schools can provide more insight to public health officials and allow students to play a bigger part in their own communities through action-mediated methods.

### 4.1. Limitations and Gaps

This review does bring to light various CS programs that have been implemented at schools positively; however, a possible limitation is that not all these programs were captured during the search, revealing potential reporting bias. First, projects may have been published before the term ‘citizen science’ was popularized. Second, it’s probable that only successful CS projects at schools become published [83].

In the realm of public health, it is possible that similar programs use similar CS approaches are actually published under PAR. While CS and PAR seem similar, PAR projects aim to complete an action that will bring some form of social change to a community by exposing a problem, constructing intervention strategies and creating a solution; CS utilizes the same methodology but without the focus on a call to action [84,85]. CS practices also exist in other activities such as popular epidemiology. While popular epidemiology is a step towards involving citizens with public health issues, popular epidemiology and CS methods have been used sparingly for infectious disease surveillance. For surveillance, some studies have utilized CS for vector-borne disease surveillance, specifically for mosquitoes and ticks, to monitor vector abundances and distributions to predict potential viral spread [40,59,60,86,87]. Beyond surveillance, CS in public health is uncommon. In an overview by the European Commission, there were no examples of public health-related CS projects; the majority of well-regarded projects were focused on biology and ecology [88]. With public health sectors tackling continued budget constraints [89] and known impacts on infectious diseases due to global climate change, the need for big data from various sources for disease surveillance is crucial in creating preparedness and response plans [90].

Based on the results of this literature review, we have identified three clear research gaps. First, none of the projects in the inventory assessed the differences between data collected between different educational levels and/or ages. Even though some of the projects were implemented at various educational levels, data quality was solely compared with expert data [38,91], rather than between different education levels as well. This lack of information regarding different citizens makes it more difficult to judge students’ abilities, recruit participants and create protocols based on different age groups. Second, there is no distinct standard measure of student or citizen engagement in CS projects in general. Participation and knowledge are not always used to measure student engagement, and in some cases, engagement was not measured at all. If some CS projects aim to also enhance scientific literacy and confidence, an accepted standard measurement is necessary to prove the concept. This is even more important when CS is focused on scholars. The third gap is project location: Most of the school CS projects in this review are from North America and Europe, in countries with high human development indexes (HDIs) [92]. With the rising popularity of CS, the untapped potential of using CS in African and Asian countries is evident [93,94,95,96]. There is even more untapped potential in Africa and Asia by using CS at school-level as an educational tool. The United Nations recognizes that education is a transformative instrument in evolving societal norms and policies [92]. Applying CS at schools in countries with lower HDIs can be a catalyst not only for scientific development but also for societal empowerment and advancement.

### 4.2. Strategies to Implement Effective School CS Programs in Infectious Disease Surveillance

(1)Consider program participants: Student participants are different from adult participants in a CS program. Developing programs that account for students’ motivations, scientific curiosities and capabilities are crucial for a program’s success.(2)Support current school curriculum and initiatives: CS projects that align well with teachers’ lesson plans and standards make implementing a CS project less demanding.(3)Create simple and clear protocols: Students focus on following procedures. However, protocols should be explained plainly and easy to follow. In addition, data collection should be accessible.(4)Take advantage of appropriate technology: Using technology that is portable, such as smartphones, can make implementing CS projects a fast process. This can also result in rapid data collection.(5)Maintain open communication and feedback with students and teachers: Students should understand their role as citizen scientists. Students and teachers need to know why they are collecting data and why they are doing so in a specific way. Discussing the impact of their work and how the data will be analyzed is also valuable.(6)Promote community outreach: CS is a community-driven scientific initiative. Involving students and their community-members enhances scientific confidence and strengthens civic cooperation.(7)Spread knowledge gained through experience and results: Publicizing CS projects and showing collaborations between experts and non-experts can build the public’s trust in science and combat misinformation.

## 5. Conclusions

Infectious diseases constitute global public health threats, and the need to prevent their spread and transmission is crucial. Strategies for early detection of infectious diseases are still in its infancy, and the COVI-19 pandemic illustrates the need for more strategies. As illustrated in this critical review, CS provides opportunities for data collection and early surveillance of emergent infectious diseases. More countries have begun to see the significance of CS, such as in the European Union: the SOCIENTIZE Consortium recognizes the possibilities of CS and the benefits to engage community members, including youth, with scientists [88]. Implementing CS programs at schools is achievable and effective, benefitting researchers and students alike. However, infectious disease surveillance through school CS initiatives is still nonexistent worldwide or remains unpublished. Students can collect data that are not always easily accessible to expert researchers, which can give a better picture of the true epidemiological situation of various diseases. Complementing student-collected data to government-led disease surveillance and other health databases could help prevent future epidemics by truly translating public engagement into public health. In conclusion, school CS programs are untapped drivers to fill in knowledge gaps not only for research advancement but for public health education and engagement and infectious disease surveillance.

## Figures and Tables

**Figure 1 ijerph-18-07019-f001:**
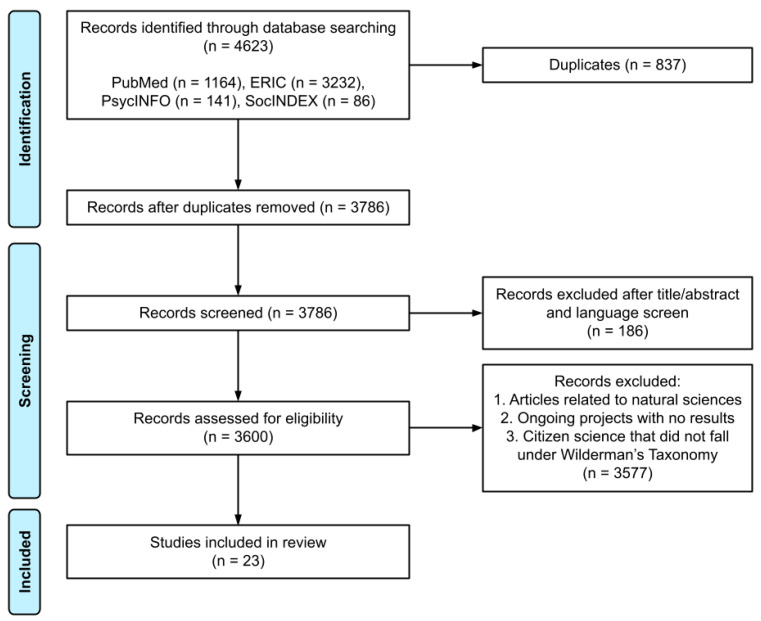
Flowchart of the selection process according to PRISMA.

**Figure 2 ijerph-18-07019-f002:**
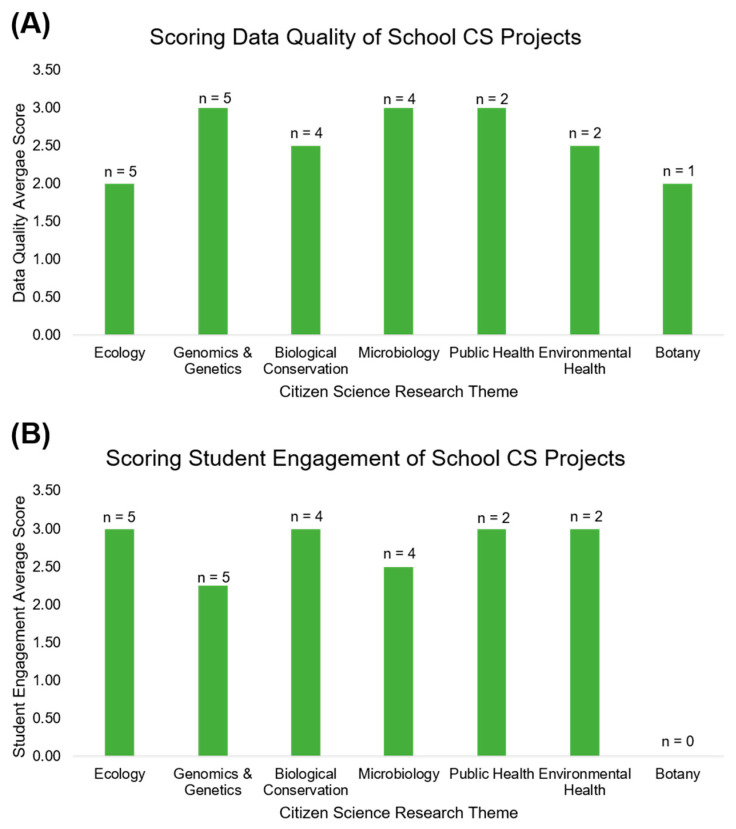
(**A**) Bar chart of the average scores for data quality in each citizen science research theme. (**B**) Bar chart of the average scores for student engagement in each citizen science research theme. Scores were from 0–3 (not mentioned, low, moderate, high).

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
