# Peer review of "The Potential Role of School Citizen Science Programs in Infectious Disease Surveillance: A Critical Review"

_ijerph, 2021, doi:10.3390/ijerph18137019_

Round 1

Reviewer 1 Report

May 27th, 2021

Review for Environmental Research and Public Health

Manuscript: Turning Students Into Amateur Researchers: a Review of Life Citizen Science Programs From Educational Centers.

Authors: Abourashed et al.

This paper aimed to review and to investigate school CS programs in the areas of (applied) life sciences with interest on assessing if any projects target infectious disease surveillance. This could agree very well with the aims of this journal. However, the paper suffers from a Major Drawbacks related with the focus/organization of the manuscript and I recommend a Major Revision and Resubmitting before acceptance.

How this review is connected with the problem of emergent infectious diseases is not clear for readers (me). Neither is clear for me why the authors are applying here the format of a research paper when they are producing a Review.  

Results (in a research paper) cannot be just the detailed descriptions of previous scholar CS programs (already explained in tables). You need analyses, comparisons, data, and figures from your analyses (besides the tables you already included).

If what I pointed out above was the real target of this manuscript (Scholar CS programs focusing on surveillance for infectious diseases) then the title and the structure of the manuscript is out of the focus.

Some proposals for titles can be:

Lessons from school citizen science programs and its potential role in efficient infectious diseases surveillance.

or

Can really school citizen science programs be a useful tool in the surveillance for emergent infectious diseases?

or

The potential role of school citizen science programs in the surveillance for emergent infectious diseases.

Sections (something like this or similar):

  • The current dilemma of the emergent infectious diseases and the need for early detection strategies.
  • Citizen Science programs as new research tools/strategies and Scholar citizen science programs: What have we learn? Success factors.
  • Guidelines/strategies to implement and promote effective scholar CS programs for the surveillance of infection diseases.
  • Conclusions

In summary, you need to describe the hot topicality of infectious diseases, the relevance of the problem (more in the last two years), and the necessity of early detection strategies. How citizen science had helped to solve other related scientific problems, Why school programs are a particular case of CS that could contribute more (and/or) different from other CS programs (added benefits: i.e.: educating/engaging young people is an asset for the future, cheap and effective in under-developed geographic areas, etc). Explain why it could be a useful tool to help in this problem (diseases surveillance).

Are there things from typical CS programs (useful for diseases detections) from which we can learn when establishing scholar CS programs??

Review previous school programs, propose indicators that could help to make valid and informative the scholar CS programs, check those parameters in those studies, Compare them (not only within Scholar CS but also among SC programs), define a list of gaps/limitations (this is partially done here).

Applied this knowledge to plan upcoming/future scholar CS program (guidelines) focusing on surveillance of the emergent infectious diseases.

Conclude.

About the evaluation criteria (to learn from it) the authors try to determine “success factors” in terms only of data quality and student engagement in the 23 projects that were assessed in their review. Much more informative items can be added …

In my opinion (and in the current version) the manuscript is not a research paper and it’s not a review, either.

The authors need to revise and redefine their real target to produce something useful.

Reviewer 2 Report

Authors,

Your paper presents an interesting premise around the opportunity of students to participate in infectious disease surveillance citizen science. However, the bulk of your paper does not seem to actually focus on this. Your paper reads a bit disjointed, as you introduce the idea of CS infectious disease surveillance but then your actual research focuses on life science education programs have incorporated CS.

Further, it’s unclear in your introduction why it is important to know the extent of infectious disease surveillance CS programs at the school level. This need was not well established in your introduction. It almost reads as an aside to the bulk of the paper.

It would be important to establish how you are defining engagement. You discuss evaluating student engagement as a key component of your analysis, but it is unclear what exactly you mean by engagement.

In terms of your methods, it seems to be a direct conflict in your methods and the expectations of your results. You mention in your screening processes you applied Wilderman’s taxonomy, however this seems to be antithetical to finding papers around student engagement, motivation, science literacy outcomes. Many of these types of social/psychological CS outcomes are often reported separately to details of the operation or data collection efficacy of citizen science programs. If you are hoping to report on these metrics, as you note in your discussion/conclusions, it would make more sense that you are specifically screening for these types of outcomes rather than just CS operational description papers.

In your discussion you say in line 399-400 that CD projects go unpublished because they are not meant to be designed for direct contribution to science. That again, is opposite to the definition to CS, even the definition you outlined in your introduction. If there are science data collection programs not designed to be contributing to genuine scientific pursuits, there are a number of different terms for that enterprise (CUREs being one) but I would not include that in CS.

Additionally, you assert in line 428-429 that a main purpose of CS is enhancing scientific literacy. Again, this might be something that is a goal of some CS projects, but it is certainly not true of all projects.

I think this paper shows a lot of promise, but would benefit from some realignment and clarity on what the intention is of this paper.

Reviewer 3 Report

The article presents a literature review on citizen science experiences at school level in life sciences. The article follows Wilderman's taxonomy to determine the moment of involvement of the participants in the collected articles. It evaluates the quality of the data obtained through the participation and the engagement of the participants with the projects for the articles found. 

The search was carried out using only the term "Citizen science". Articles that could have been included in this literature review have likely been excluded as they may have used other terminology. However, the authors have already noted this limitation in the discussion.

The inclusion of the assessment of citizen science for infectious disease surveillance lacks a connection to the rest of the article.  The methodology does not seem to be focused on an assessment or possible application of citizen science in that area. It said that no citizen science experience related to it has been found, but neither about many other possible applications of citizen science. However, it seems to be the main theme in the conclusions. 

I recommend a minor revision of this article based on the above.

Round 2

Reviewer 1 Report

Im completely agree with the changes made by the authors changes on their manuscript. Thanks for it. However, the abstract has not been modified and then its lacking there all the improvements made on the MS.

Please, rewrite it and incorporated briefly, at the end of the abstract, the main conclusions, fundamental gaps, and lessons/main recommendations you discovered to guide people when using CS projects as a tool for diseases surveillance. 
